# Interactions with RNA direct the Polycomb group protein SCML2 to chromatin where it represses target genes

Roberto Bonasio[1†‡], Emilio Lecona[1†§], Varun Narendra[1], Philipp Voigt[1], Fabio Parisi[2,3], Yuval Kluger[2,3], Danny Reinberg[1]*

[1]Department of Biochemistry and Molecular Pharmacology, Howard Hughes Medical Institute, New York University School of Medicine, New York, United States; [2]Department of Pathology, Yale University School of Medicine, New Haven, United States; [3]Yale Cancer Center, Yale University School of Medicine, New Haven, United States

*For correspondence: danny.reinberg@nyumc.org

[†]These authors contributed equally to this work

Present address: [‡]Department of Cell and Developmental Biology, Perelman School of Medicine, University of Pennsylvania, Philadelphia, United States; [§]Genome Instability Group, Spanish National Cancer Research Centre (CNIO), Madrid, Spain

**Abstract** *Polycomb* repressive complex-1 (PRC1) is essential for the epigenetic regulation of gene expression. SCML2 is a mammalian homolog of *Drosophila* SCM, a *Polycomb*-group protein that associates with PRC1. In this study, we show that SCML2A, an SCML2 isoform tightly associated to chromatin, contributes to PRC1 localization and also directly enforces repression of certain *Polycomb* target genes. SCML2A binds to PRC1 via its SPM domain and interacts with ncRNAs through a novel RNA-binding region (RBR). Targeting of SCML2A to chromatin involves the coordinated action of the MBT domains, RNA binding, and interaction with PRC1 through the SPM domain. Deletion of the RBR reduces the occupancy of SCML2A at target genes and overexpression of a mutant SCML2A lacking the RBR causes defects in PRC1 recruitment. These observations point to a role for ncRNAs in regulating SCML2 function and suggest that SCML2 participates in the epigenetic control of transcription directly and in cooperation with PRC1.

## Introduction

Polycomb group (PcG) genes are key epigenetic regulators in *Drosophila*, mammals, and plants (*Schuettengruber et al., 2007*). PcG mutants fail to maintain transcriptional repression of differentiation genes during development, which results in their ectopic expression, compromised patterning of the embryo, and homeotic transformations (*Lewis, 1978*; *Struhl, 1981*; *Simon et al., 1992*). Most PcG genes encode proteins that recognize chromatin or modify its structure, often by forming multi-subunit protein complexes such as *Polycomb* repressive complex 1 (PRC1) (*Simon and Kingston, 2009*). Genetic screens and biochemical studies have identified several additional PcG proteins that are not stable components of known *Polycomb* complexes. One of these proteins is sex comb on midleg (SCM), which is required for appropriate body patterning during embryonic development in *Drosophila* (*Simon et al., 1992*; *Bornemann et al., 1996*, *1998*). SCM and its human homologs were recovered in sub-stoichiometric amounts after biochemical purifications of the PRC1 complex (*Shao et al., 1999*; *Levine et al., 2002*; *Gao et al., 2012*), suggesting a functional link between SCM and PRC1. SCM also belongs to a small family of proteins characterized by the presence of a conserved malignant brain tumor domain (MBT), which functions as a binding module for methylated lysines on histones (*Bonasio et al., 2010a*), further suggesting that SCM might function on chromatin. In addition to the MBT domain, SCM contains a C-terminal SPM domain, through which it is thought to interact with PRC1 (*Peterson et al., 2004*).

**eLife digest** Almost every cell in our bodies has the same genes but at any one time different genes will be switched on in different cells. Much of the DNA in a cell is wrapped around proteins called histones to form a compact structure called chromatin. The DNA in chromatin is often so tightly packed that the genes in the DNA cannot to be accessed or switched on.

A complex of proteins called the Polycomb Repressive Complex 1 (or PRC1 for short) pack DNA into chromatin to switch genes off (and keep them off) in both plants and animals. Other proteins are known to weakly bind to this complex, including one called SCM in fruit flies. However, this protein has not been extensively studied.

Bonasio, Lecona et al. have now looked at a related version of this protein that binds to chromatin in humans. These experiments revealed that this protein, which is called SCML2A, also binds to molecules of RNA, and Bonasio, Lecona et al. also identified a previously unrecognized domain within SCML2A that interacts with these molecules.

Like other domains in this protein that bind to PRC1 and histones, SCML2A needs its RNA-binding region to be able to bind to chromatin in order to target and switch off certain genes. As such, the findings of Bonasio, Lecona et al. support models whereby RNA molecules can regulate the expression of genes, and suggest that some RNA molecules do this by interacting with SCML2A.

Future work is needed to address whether RNA molecules serve as guides that direct SCML2A to specific genes that need to be switched off, or whether these molecules' roles are more complex.

Despite the crucial importance of PcG proteins in gene regulation, development, and disease (*Sparmann and van Lohuizen, 2006*), a coherent model that explains how these proteins are recruited and regulate different target genes in different cell lineages is lacking (*Simon and Kingston, 2009*). In *Drosophila*, arrays of DNA motifs recognized by a variety of transcription factors function as recruitment 'hubs' for PRCs and are known as *Polycomb* responsive elements (PREs) (*Müller and Kassis, 2006*); however, PREs alone are not sufficient to explain or predict the genome-wide occupancy patterns of PcG complexes (*Simon and Kingston, 2009*). In mammals, our understanding of PRC targeting is even more limited, given that only a few examples of PRE-like DNA elements have been described to date (*Bengani et al., 2013*; *Cuddapah et al., 2012*; *Sing et al., 2009*; *Woo et al., 2010*, *2013*) and several of the transcription factors that bind and recruit *Drosophila* PRCs are absent or poorly conserved (*Schuettengruber et al., 2007*). Moreover, there are several mammalian versions of the two best characterized *Drosophila* PcG complexes, PRC1 and PRC2, which differ in subunit composition and chromatin localization (*Kuzmichev et al., 2004*; *Margueron et al., 2008*; *Gao et al., 2012*).

The mammalian PRC1 closest in subunit composition to their *Drosophila* homolog (PRC1.2 and PRC1.4 [*Gao et al., 2012*]) have conserved functions in development and disease (*Richly et al., 2011*) and contain subunits that recognize trimethylated lysine 27 of histone H3 (H3K27me3), the catalytic product of PRC2 (*Cao et al., 2002*; *Kuzmichev et al., 2002*). However, the mechanistic details of PRC1 repression in mammals are poorly understood. Although H3K27me3 may contribute to stabilizing some forms of PRC1 on chromatin, the genome-wide distribution of PRC1 is largely unchanged in embryonic stem cells that do not have H3K27me3 (*Tavares et al., 2012*), and many PRC1 complexes do not contain the subunit that binds to H3K27me3 (*Gao et al., 2012*). Various factors, including RUNX1 (*Yu et al., 2012*) and the RNA helicase MOV10 (*El Messaoudi-Aubert et al., 2010*), have also been proposed to guide PRC1 recruitment in specific contexts, but an overarching model remains lacking. Given that increasing evidence points to a role for noncoding RNAs (ncRNAs) in mediating recruitment of chromatin complexes (*Koziol and Rinn, 2010*; *Wang and Chang, 2011*) and that at least one ncRNA has been shown to make contact with PRC1 (*Yap et al., 2010*), we hypothesized that an RNA-binding chromatin protein may be involved in PRC1 recruitment in mammals.

The potential role of SCM and its mammalian homologs in the regulation of PRC1 and function remain unexplored. A recent study showed that *Drosophila* SCM is recruited to the PRE upstream of *Ultrabithorax* independent of PRC1 or PRC2, whereas PRC1 and PRC2 binding requires the presence of SCM (*Wang et al., 2010b*). The human genome comprises four genes with homology to *Scm*, but only two that contain conserved MBT, SPM, and DUF3588 domains, *SCMH1* and *SCML2*. *SCMH1* functions in spermatogenesis (*Takada et al., 2007*), whereas little is known about *SCML2*, except that

it is ubiquitously expressed and that deletions of its coding sequence are found in a subset of medulloblastomas (*Northcott et al., 2009*), suggesting it may have tumor-suppressive activity.

In this study, we report that SCML2 interacts with ncRNAs through an RNA-binding region (RBR), contributes to the recruitment of PRC1 to target genes and also directly collaborates in gene repression. Furthermore, we show that the chromatin localization of SCML2 is achieved through the combined action of the RBR, MBT, and SPM domains and that the expression of ΔRBR mutants of SCML2 causes mislocalization of PRC1.

## Results

### SCML2 interacts with RNA through a novel RNA-binding region

Interaction with RNA regulates recruitment and assembly of several chromatin-associated factors, including members of the *Polycomb* and *trithorax* groups (*Rinn et al., 2007*; *Khalil et al., 2009*; *Kanhere et al., 2010*; *Tsai et al., 2010*; *Wang et al., 2011*; *Kaneko et al., 2013, 2014*). To find novel RNA-interacting proteins among potential epigenetic regulators, we performed a candidate-based screen using RNA immunoprecipitation (RIP) from HeLa S3 nuclear extracts. We observed that the PcG protein SCML2, a homolog of *Drosophila* SCM, consistently co-purified with a relatively large amount of RNA, especially when compared to EZH2 and MLL, which are part of complexes known to interact with ncRNAs (*Rinn et al., 2007*; *Kaneko et al., 2010, 2013*; *Wang et al., 2011*; *Figure 1A*). SCML2 did not co-precipitate with either EZH2 or MLL or vice versa (data not shown).

RNA binding assays with total HeLa S3 RNA and recombinant fragments of SCML2 (*Figure 1B*) revealed that amino acids 256–330 were necessary and sufficient for RNA binding in vitro (*Figure 1C*). Although the most prominent bands recovered by pull-down of SCML2 fragments corresponded to ribosomal RNA, this was most likely due to their abundance in the input and not to a specific interaction. Other chromatin-associated proteins that bind RNA are rather promiscuous in vitro (*Davidovich et al., 2013*; *Kaneko et al., 2014*); therefore, this type of assay is only suitable to identify the protein region that harbors an affinity for RNA. Because the region spanning 256–330 bears no resemblance to other known RNA-binding domains, we tentatively refer to it as the RNA binding region (RBR) of SCML2. The SCML2 RBR exhibited little specificity for RNA sequence, but discriminated between RNA and DNA in vitro (*Figure 1D*), and strongly preferred single-stranded to double-stranded RNA (*Figure 1D*, left). SCML2 fragments lacking the RBR (ΔRBR) failed to bind any of the nucleic acids tested (*Figure 1D*, right).

### The RBR is sufficient for RNA binding in vitro

To obtain a more quantitative description of the nucleic acid binding preferences of SCML2, we performed electromobility shift assays (EMSAs) with a recombinant SCML2 fragment encompassing the RBR alone (aa 256–330). The RBR shifted both *HOTAIR* RNA and DNA (*Figure 2A*), and so did the larger MBT-DUF fragment (*Figure 2—figure supplement 1A*), but only *HOTAIR* RNA and not DNA competed efficiently for binding to the RBR (*Figure 2B*) or to the MBT-DUF fragments (*Figure 2—figure supplement 1B*).

Because a basic patch was reported to mediate SCML2–DNA and SCML2–nucleosome interactions in vitro (*Nady et al., 2012*) and partially overlaps with the RBR, we also tested the ability of this fragment to bind nucleosomes by EMSA. The RBR displayed an affinity for nucleosome particles (*Figure 2A*), but much lower than for RNA as demonstrated by the fact that nucleosomes could not compete with *HOTAIR* RNA for binding to the RBR (*Figure 2B*). When both nucleosomes and RNA were present in limiting amounts, we noticed the appearance of a super-shifted band that was not observed when either species were added separately (*Figure 2C*, asterisk), suggesting that the RBR—and therefore SCML2—is capable of forming a ternary complex with nucleosomes and RNA simultaneously (*Figure 2—figure supplement 2A*). The appearance of the super-shifted band was not due to GST-mediated dimerization, as it also occurred after the removal of the GST moiety from the RBR (*Figure 2—figure supplement 2B*). The possibility of ternary complex formation is further supported by the observation that nucleosomes co-precipitated in vitro with both the SCML2 RBR and HOTAIR simultaneously (*Figure 2—figure supplement 2C*), whereas HOTAIR alone did not bind nucleosomes (*Figure 2—figure supplement 2D*).

Not all RNA sequences were bound by the RBR to the same extent. In vitro-transcribed RNA from the 601 nucleosome-positioning template competed less efficiently for binding to the RBR as compared to HOTAIR (*Figure 2B*). We also tested the binding specificity of the SCML2 RBR for isolated stem-loops within the 5′ fragment of the long noncoding RNA (lncRNA) *HOTAIR* that interacts with

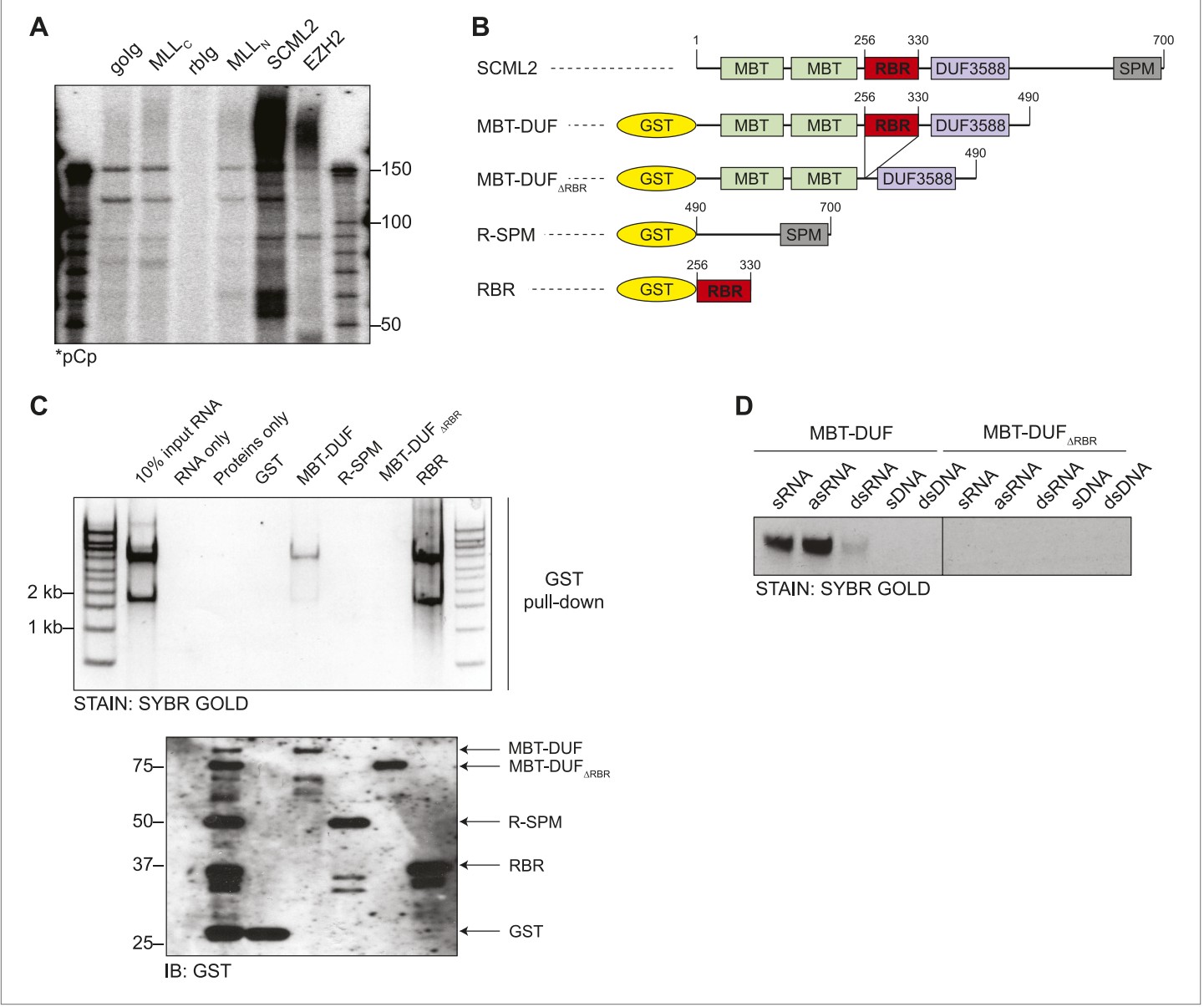

**Figure 1**. Identification of the RNA-binding region of SCML2. (**A**) Endogenous RIP from HeLaS3 nuclear extract for the indicated chromatin proteins or with goat (goIg) and rabbit (rbIg) control antibodies. Co-precipitating RNA was labeled in 3' and resolved by denaturing PAGE. (**B**) Domain organization of SCML2 and scheme of the GST-fused truncations. (**C**) In vitro pull-down of total RNA with the indicated SCML2 fragments. RNA fraction (top) and protein fraction (bottom) are shown. (**D**) The indicated sense RNA (sRNA) and antisense RNA (asRNA) were transcribed in vitro from a synthetic random 100 nts DNA template, PAGE-purified and annealed to form double-stranded RNA (dsRNA). The sense DNA oligonucleotide (sDNA) and the double-stranded DNA template (dsDNA) were used as controls.

PRC2 (*Tsai et al., 2010*). Only the full 1–300 fragment bound to the SCML2 RBR with sufficient affinity to co-precipitate, whereas none of the isolated stem-loops did (data not shown), suggesting that the RBR is capable of recognizing complex structures and/or sequences and that it does not just form non-specific interactions with any type of RNA.

## RBR, MBT, and SPM domains coordinate recruitment of SCML2A to chromatin

Having found that SCML2 binds to RNA in vivo and in vitro, we sought to determine the consequences of these binding events on the cellular and molecular functions of SCML2.

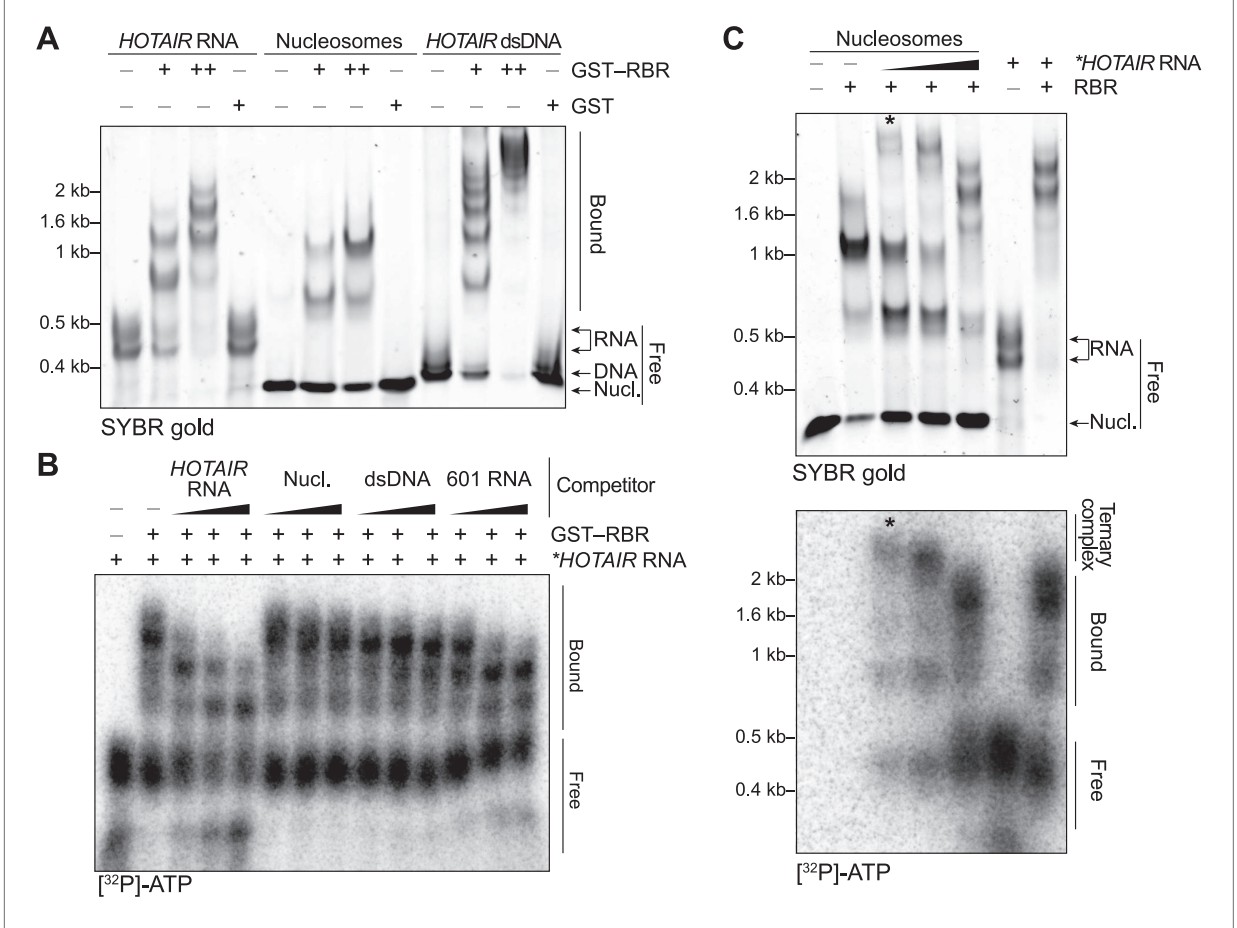

**Figure 2**. In vitro characterization of the binding preferences of the SCML2 RBR. (**A**) EMSA with 2 (+) and 4 (++) pmol GST-fused RBR and 520 fmol *HOTAIR* RNA 1–300, 690 fmol nucleosomes, or 275 fmol dsDNA encoding *HOTAIR*$_{1–300}$. 5.2 pmol GST were used as a control. Complexes were separated on native gels and detected with SYBR gold stain. Data are representative of ≥4 experiments. (**B**) EMSA with 3.4 pmol GST–RBR and 520 fmol of labeled *HOTAIR* RNA was performed as in (**A**) with the addition of, from left to right, 260, 520, and 1040 fmol of unlabeled *HOTAIR* RNA, nucleosome particles, dsDNA, and *601* RNA. Bands were visualized by autoradiography. Data are representative of ≥4 experiments. (**C**) EMSA of RBR with nucleosomes and labeled HOTAIR RNA. Assay was performed as in (**B**) varying the amount of labeled RNA. Nucleic acid stain (top) and autoradiography (bottom) are shown. The asterisk indicates the putative ternary complex.

The following figure supplements are available for figure 2:

**Figure supplement 1**. Additional EMSAs.

**Figure supplement 2**. Ternary complex EMSAs and pull-downs.

As we reported previously, the human *SCML2* locus gives rise to two alternative isoforms: one encodes the full-length version (SCML2A) and the other a C-terminal truncation, generated by alternative splicing (SCML2B; *Figure 3—figure supplement 1A*) (*Lecona et al., 2013*). SCML2A has a domain organization similar to that of *Drosophila* SCM: two N-terminal MBT repeats, a DUF3588 domain, and a C-terminal SPM domain. Although both SCML2 isoforms are predominantly nuclear, SCML2A is enriched in the chromatin fraction, whereas the bulk of SCML2B, which lacks the SPM domain is found in the soluble nuclear fraction, where it interacts with the cell cycle machinery and participates in the G1/S checkpoint (*Lecona et al., 2013*). Therefore, SCML2A is the isoform more likely to have a direct function on chromatin and transcription and is the focus of the remainder of this study.

Multiple lines of evidence point to a role for ncRNAs in recruiting epigenetic regulators, in particular members of the PcG and trxG, to chromatin (*Rinn et al., 2007*; *Koziol and Rinn, 2010*; *Bonasio et al.,*

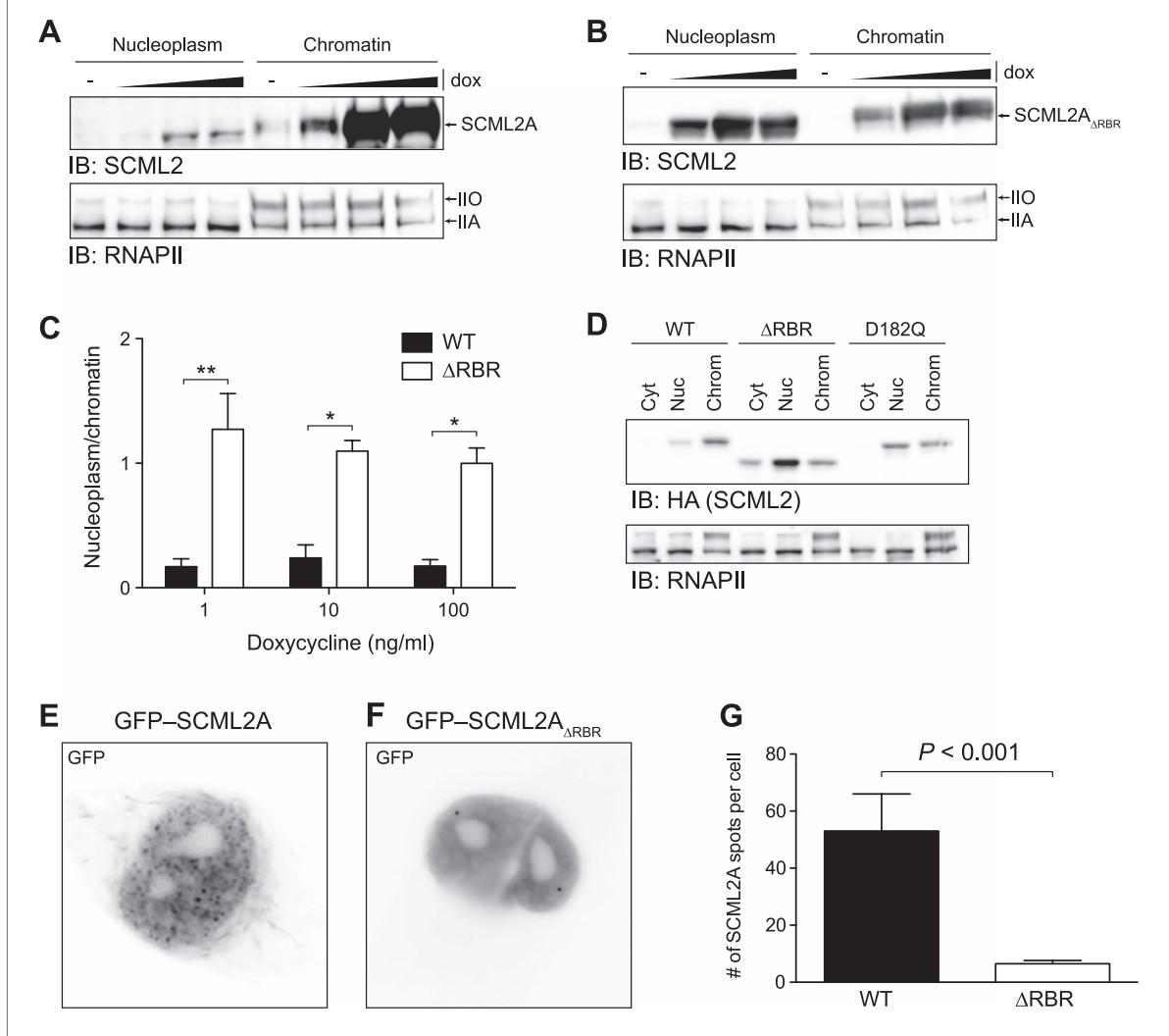

**Figure 3**. Loss of the RBR alters the subnuclear distribution of SCML2A. (**A** and **B**) Salt fractionation of 293T-REx induced to express SCML2A WT (**A**) and ΔRBR (**B**) with increasing amounts of doxycycline (dox). The distribution of RNAPIIA and RNAPIIO are shown as fractionation and loading control. (**C**) Densitometry-based quantification of the ratio of nucleoplasmic vs chromatin-associated SCML2A WT (black bars) and ΔRBR (white bars). Bars show mean measurements from two independent clones at increasing levels of inductions. p-values were calculated with 2-way ANOVA. (**D**) Salt fractionation of 293T-REx induced to express SCML2A WT, ΔRBR, and the MBT-domain mutant D182Q. (**E** and **F**) Fluorescence microscopy of HeLa cells transiently transfected with vectors encoding GFP-SCML2A_WT (**E**) or GFP-SCML2A_ΔRBR (**F**). (**G**) Quantification of SCML2 spots in multiple cells transfected as in (**E**) and (**F**). Bars represent mean + SEM. $N = 11$ for SCML2A; $N = 17$ for SCML2A_ΔRBR. p-value was calculated with an unpaired $t$ test.

The following figure supplements are available for figure 3:

**Figure supplement 1**. Nuclear distribution of SCML2A_D182A and SCML2B.

2010b; *Wang et al., 2011*). Because SCML2A exhibited stronger affinity for RNA than for DNA, we hypothesized that interactions between RNA and the RBR may contribute to the recruitment of SCML2 to chromatin. Consistent with this hypothesis, an SCML2A mutant lacking the RBR (SCML2A_ΔRBR) was displaced from chromatin, as demonstrated by the fact that a larger fraction of this protein was released with low-salt extraction when compared to the wild-type (WT) form, regardless of the amount of transgenic protein being expressed (*Figure 3A–C*).

We then assessed the potential role of the MBT domains in binding to chromatin. Mutation of the acidic residue that coordinates the methylated lysine within the aromatic cage impairs its binding (*Sathyamurthy et al., 2003; Grimm et al., 2007; Santiveri et al., 2008*) and also resulted in loss of SCML2A from the chromatin fraction, but not to the extent observed with the RBR deletion (*Figure 3D*,

*Figure 3—figure supplement 1B*). The SCML2 isoform lacking the SPM domain, SCML2B, displayed a predominantly diffuse distribution within the nucleus, which was not affected by the removal of the RBR; however the residual amount of SCML2B that could be recovered from the chromatin fraction was further decreased upon deletion of the RBR (*Figure 3—figure supplement 1C–G*). Together, these results show that the affinity of SCML2 for chromatin is regulated by multiple molecular mechanisms, including binding to PRC1 via the SPM domain, histone marks via the MBT domains, and RNA via the RBR.

We confirmed the altered distribution of SCML2A$_{\Delta RBR}$ by fluorescence microscopy. Similar to the endogenous SCML2 protein (*Figure 3—figure supplement 1H*), GFP–SCML2A accumulated in the nucleus and exhibited a punctate distribution, suggestive of recruitment to specific sub-nuclear compartments, presumably target sites on chromatin (*Figure 3E*). In contrast, GFP-SCML2A$_{\Delta RBR}$ exhibited a diffuse nuclear distribution (*Figure 3F*) consistent with its looser association with chromatin revealed by biochemical fractionations (*Figure 3A–C*). The number of intense GFP spots decreased from a median value of 39 per cell for SCML2A to only 5 per cell for SCML2A$_{\Delta RBR}$ (*Figure 3G*). Deletion of the RBR did not affect the ability of SCML2A to be imported into the nucleus, nor its ability to interact with other protein factors, including the core subunits of the PRC1 complex (see below).

## SCML2 interacts with PRC1 and represses transcription

To further explore the functional relationship between SCML2 and PRC1, we purified epitope-tagged SCML2A and SCML2B from the chromatin fraction of 293T-REx cells. Both SCML2A$_{WT}$ and SCML2A$_{\Delta RBR}$ co-purified with all the components of the canonical PRC1, including RING1A, RING1B, CBX2/8, PHC2/3, and BMI1/PCGF4 (a subunit specific for PRC1.4 [*Gao et al., 2012*]), suggesting that deletion of the RBR does not affect the ability of SCML2A to interact with PRC1. Only small traces of PRC1 components could be detected among the SCML2B-associated proteins, despite the fact that a comparable number of SCML2 peptides were detected in the two purifications (*Figure 4A*). SCML2A also co-purified with SCMH1 (the other SCM homolog in humans) and L3MBTL3, both of which comprise C-terminal SPM domains (*Figure 4A*). As a control, both SCML2A and SCML2B recovered comparable amounts of USP7, in good agreement with previous reports (*Sowa et al., 2009*) and as we also independently confirmed (Lecona et al. unpublished). In keeping with these observations, immunoprecipitation (IP) of BMI1 from the chromatin fraction of HeLa cells recovered preferentially the chromatin-associated and SPM-containing SCML2A isoform (*Figure 4B*, compare top and bottom band in lane 4 vs lane 1 or lane 3).

Consistent with the classification of *SCML2* as a PcG (and therefore repressive) gene, artificial tethering of both SCML2A and SCML2B to an integrated reporter using the DNA-binding domain of *Saccharomyces cerevisiae* Gal4p (*Vaquero et al., 2004*) resulted in a dose-dependent repression of transcription (*Figure 4C*). However, only tethering of SCML2A but not SCML2B caused an increase in BMI1 recruitment to the *UAS*-containing reporter (*Figure 4—figure supplement 1A,B*), confirming that the SPM domain is required for interacting with BMI1 and suggesting that SCML2A contributes to PRC1 targeting. On the other hand, recruitment of BMI1 was not necessary for SCML2-mediated repression in this context. Consistent with this observation, both the MBT domains and the SPM domain contribute to the repressive activity of SCM in *Drosophila* (*Roseman et al., 2001*; *Peterson et al., 2004*; *Grimm et al., 2007*).

These results show that RNA binding and the interaction with PRC1 are independent events and that they require different protein regions (the RBR and SPM domain, respectively). Our data support a model in which the RBR contributes to the localization of SCML2A to chromatin, where it interacts with PRC1 and creates a chromatin environment that is refractory to transcription.

## SCML2 is enriched at PRC target sites on chromatin

Because SCM is a PcG protein and binds to at least one PcG target in *Drosophila* (*Wang et al., 2010b*), we reasoned that we could gain further insight on the function of SCML2 and its relationship with the *Polycomb* pathway by determining its genome-wide localization using ChIP sequencing (ChIP-seq). Following a strategy we previously described (*Gao et al., 2012*; *Kaneko et al., 2013*), we generated 293T-REx lines carrying stably integrated, inducible transgenes for human SCML2 fused to a triple N-terminal affinity tag (FLAG-HA-Twin-Strep-tag, henceforth 'N3') and shRNAs against the endogenous *SCML2*. In these cells, more than 30% (781 out of 2479 total) of SCML2-enriched regions (ERs) were located in the vicinity (<10 kb) of the transcription start site (TSS) of protein-coding genes (*Figure 4D*),

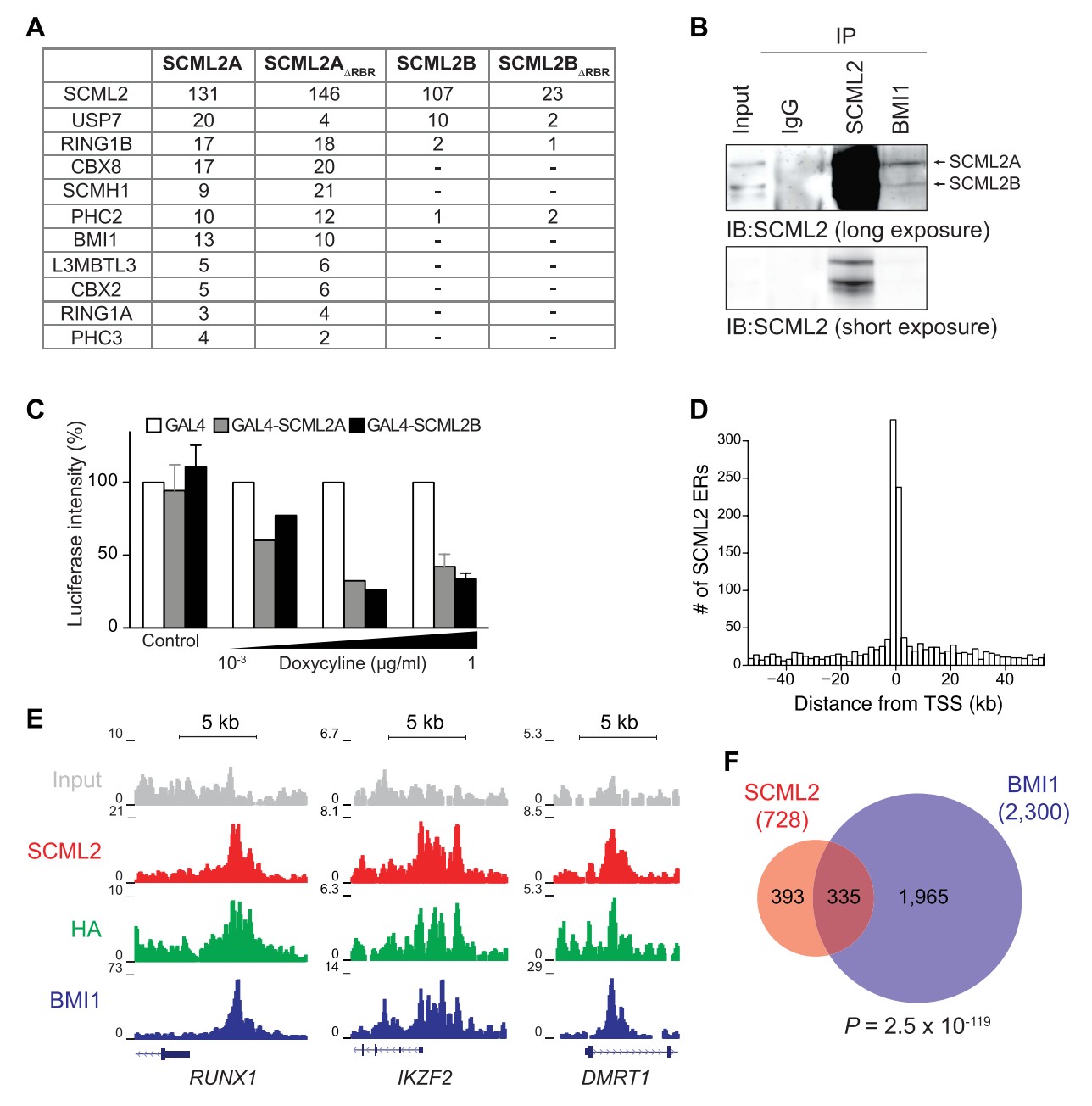

**Figure 4**. SCML2 interacts and shares target genes with BMI1. (**A**) Proteins detected by mass spectrometry after affinity purification of SCML2A, SCML2A$_{\Delta RBR}$, SCML2B and SCML2B$_{\Delta RBR}$ from the chromatin fraction of 293T-REx cells. The number of peptides identified for each protein is indicated. (**B**) IP of SCML2 and BMI1 from the chromatin fraction of HeLa cells. Two exposures are shown. (**C**) 293T-REx cells containing an integrated luciferase reporter preceded by UAS repeats were induced to express the indicated transgenes with increasing amounts of doxycycline. 24 hr after induction the activity of the luciferase reporter was assayed. Bars represent the luciferase activity normalized to the non-induced GAL4-only control. (**D**) Distribution of SCML2 ERs around TSSs of protein coding genes. (**E**) Normalized ChIP-seq read densities for SCML2 (red), HA (green), and BMI1 (blue) as compared to input (gray) at three representative loci. (**F**) Overlap of gene targets for SCML2 and BMI1, as defined by having an ER for the respective protein within 10 kb of the TSS. p-value was calculated according to the hypergeometric distribution.

The following figure supplements are available for figure 4:

**Figure supplement 1**. Artificial recruitment of SCML2 and ChIP-seq analysis.

consistent with a role for SCML2 in gene regulation. SCML2 targets were significantly enriched for gene ontology (GO) terms related to transcriptional regulation, development, and differentiation (*Figure 4—figure supplement 1C*, *Supplementary file 1A*), as observed for targets of the *Polycomb* repressive machinery in other systems (*Schuettengruber et al., 2007*). To determine whether SCML2 shared target genes with PRC1, we also performed ChIP-seq for BMI1 (also known as PCGF4), the key component of canonical PRC1.4 (*Gao et al., 2012*). SCML2 and BMI1 co-occupied several promoters, including those of transcription factors *RUNX1*, *IKZF2*, and *DMRT1* (*Figure 4E*). Overall, SCML2 shared ~46% of its target genes with BMI1 (*Figure 4F*, *Supplementary file 1B*), almost four times more than expected by chance (p=2.5 × $10^{-119}$, hypergeometric distribution). This overlap became more pronounced as a function of decreasing p-values for the ERs (*Figure 4—figure supplement 1D,E*), suggesting that many of the SCML2-only or BMI1-only sites were false positives due to noise in the ChIP-seq signal. We made similar observations for endogenous SCML2 in non-transgenic K562 cells (*Figure 4—figure supplement 1F,G*), demonstrating that our conclusions can be generalized and do not depend on the presence of the N-terminal tag or on exogenous expression.

Together, our genome-wide analysis of SCML2 distribution on chromatin supports the notion that human SCML2, like its *Drosophila* homolog, belongs to the PcG and regulates the expression of developmental genes and other *Polycomb* targets.

## The RBR is required for efficient binding of SCML2A to a subset of PRC1-regulated loci

Next, we sought to determine whether the large-scale redistribution of SCML2A$_{\Delta RBR}$ that we observed by microscopy and biochemical fractionation (*Figure 3*) was due to the loss of recruitment of SCML2A to target genes. To this end, we compared the distribution of N3-tagged SCML2A$_{WT}$ and SCML2A$_{\Delta RBR}$ in transgenic 293T-REx cells. In the absence of doxycycline, endogenous SCML2 levels were equal in all transgenic lines and equivalent to those measured in the parental 293T-REx cell line (data not shown). Upon doxycycline treatment, the endogenous copy of SCML2 was partially silenced by the shRNA and HA-tagged SCML2A$_{WT}$ and SCML2A$_{\Delta RBR}$ were expressed at comparable levels (*Figure 5—figure supplement 1A*). Pull-down of these epitope-tagged proteins did not recover untagged SCML2 (*Figure 5—figure supplement 1B*) and could therefore be used to determine the specific localization of SCML2 WT and ΔRBR without confounding effects due to interactions with endogenous SCML2.

Visual inspection of ChIP-seq profiles revealed that SCML2$_{\Delta RBR}$ was lost from several target genes (*Figure 5A*), suggesting that the RBR, and presumably RNA–protein interactions might participate in the recruitment or stabilization of SCML2 on chromatin. A smaller number of SCML2 ERs, including the ER near the *RUNX1* TSS, exhibited SCML2A$_{\Delta RBR}$ levels comparable to WT (*Figure 5A*), consistent with the existence of RNA-independent pathways of SCML2A recruitment at some loci. Nonetheless, RBR-independent ERs constituted a small proportion of the total SCML2 binding profile, because the N3–SCML2$_{\Delta RBR}$ ChIP-seq signal was much weaker than WT when all significant (p<$10^{-5}$ according to MACS 1.4.0rc2) ERs for SCML2 WT were combined in a single meta-ER plot (*Figure 5B*).

Consistent with our observations that SCML2A colocalizes with BMI1 (*Figure 4E,F*) and can recruit it to chromatin (*Figure 4—figure supplement 1B*), partial replacement of endogenous SCML2 with SCML2A$_{\Delta RBR}$ resulted in a decreased, albeit not abolished, accumulation of BMI1 at several of the target sites (*Figure 5A,C*). The residual recruitment of BMI1 may be explained either by the fact that the knockdown of endogenous *SCML2* is far from complete (*Figure 5—figure supplement 1A*) or by the contribution of RBR-independent loci. When inspected at the level of individual genomic regions, out of 148 high-confidence ERs obtained from the triple intersection of SCML2, HA, and BMI1 ERs, a majority displayed losses in SCML2 binding in the absence of the RBR and many also had decreased BMI1 occupancy (*Figure 5E*). We confirmed these findings with two additional ChIP-seq data sets from independent biological replicates (*Figure 5—figure supplement 1C*) and ChIP-qPCR analyses of a sample of ERs (*Figure 5—figure supplement 1D*).

From this set of experiments, we conclude that SCML2 is recruited to a subset of target genes through a process that requires its RNA-binding activity and that the presence of mutant SCML2A$_{\Delta RBR}$ that is not chromatin-bound causes a partial release of BMI1 and, presumably, PRC1 from chromatin.

## Genome-wide characterization of SCML2-bound ncRNAs by RIP-seq

Given that SCML2 binds RNA through an RBR that is required for proper genome-wide targeting and downstream recruitment of PRC1, we sought to identify the ncRNAs bound to SCML2 in vivo. To this

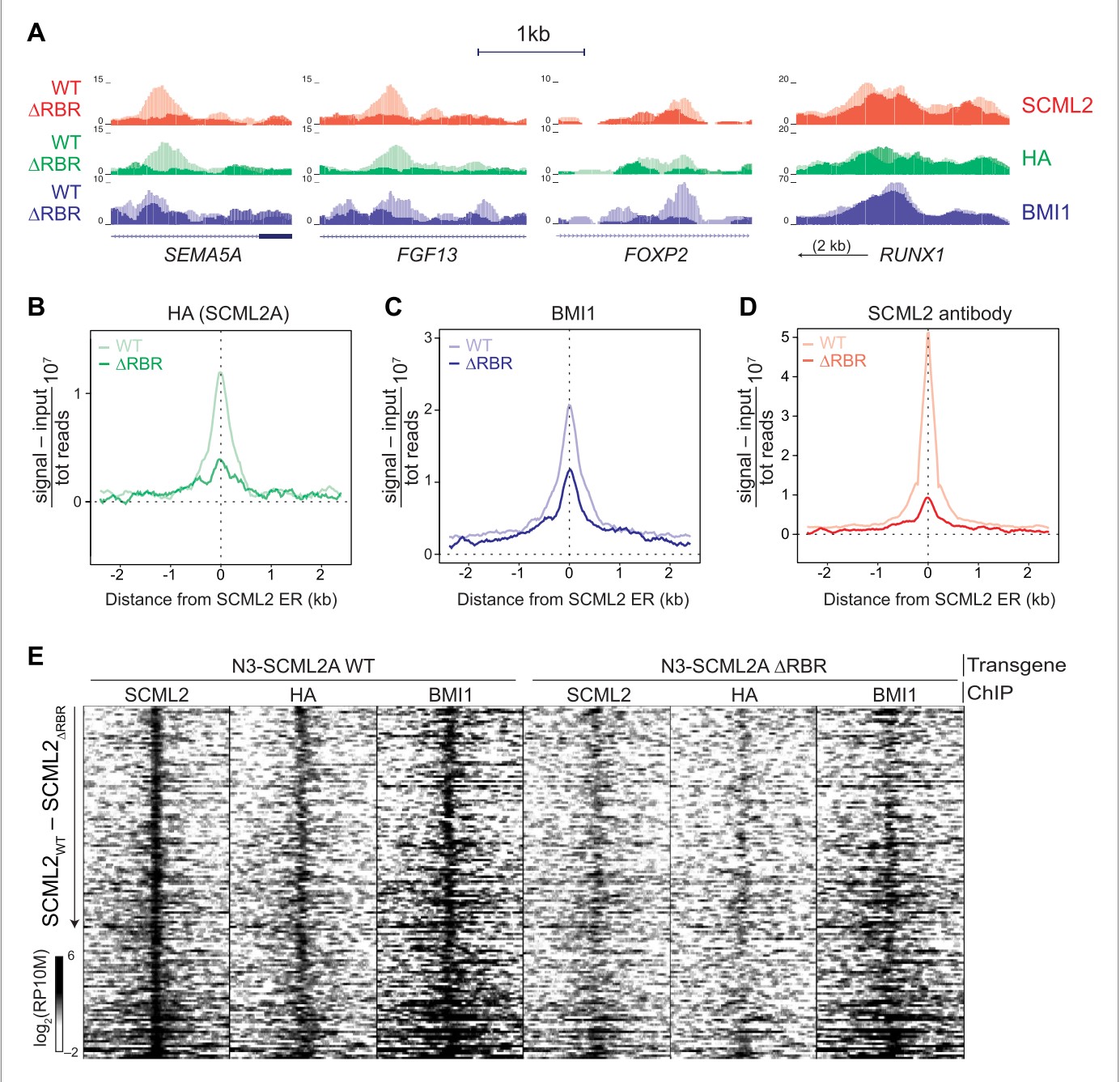

**Figure 5**. The role of the RBR in the localization of SCML2A and PRC1 to target genes. (**A**) Normalized read density profiles for SCML2 (red), HA (green), and BMI1 (blue) in cells expressing N3-SCML2A WT (light colors) or ΔRBR (dark colors). Three RBR-dependent and one RBR-independent sites are shown. (**B**–**D**) Normalized ChIP-seq enrichment (signal reads–input reads) for HA (**B**), BMI1 (**C**), and SCML2 (**D**), centered around 2479 SCML2 ERs in 293T-REx::N3-SCML2A. (**E**) Heatmap of read densities for the indicated ChIP-seq samples centered on 148 overlapping ERs for SCML2, HA, and BMI1 in the WT sample. Each row represents a genomic region spanning ±2.5 kb from the center of the ER. For each sample the log-converted normalized read density was calculated in 25 bp bins and is represented by color intensity. Regions were sorted by difference in SCML2 read densities in WT vs ΔRBR transgenic lines.

The following figure supplements are available for figure 5:

**Figure supplement 1**. Validation and replication of ChIP-seq for SCML2 WT and ΔRBR.

end, we performed HA IPs in 293T-REx expressing N3-tagged SCML2A WT, ΔRBR, or the N3 tag alone. Although comparable amounts of WT and ΔRBR protein were recovered, only in the case of WT SCML2A, we could detect co-IP of RNA (*Figure 6A,B*), which we subjected to deep sequencing (RIP followed by sequencing, RIP-seq).

Because increasing experimental evidence points to a role for long intergenic ncRNAs (lincRNAs) in the function of chromatin-associated epigenetic regulators (*Koziol and Rinn, 2010*; *Wang and Chang, 2011*), we measured the abundance of these RNAs in the IP'ed material by referring to a published lincRNA database (*Cabili et al., 2011*). Although overall abundant lincRNAs were more likely to be detected by RIP-seq, several specific interactions distinguished the set of RNAs recovered in RIPs for SCML2A WT from all other controls (*Figure 6C*), demonstrating the specificity of RNA binding by SCML2 in vivo.

Next, we tested whether the set of ncRNAs associated with SCML2 was fixed in a given cell type or could be altered by processes such as differentiation. To this end, we performed RIP-seq for endogenous SCML2 in K562 before and after treatment with hemin or PMA, which stimulate their differentiation toward the erythroid or megakaryocytic lineage, respectively (*Jacquel et al., 2006*). Indeed, cell

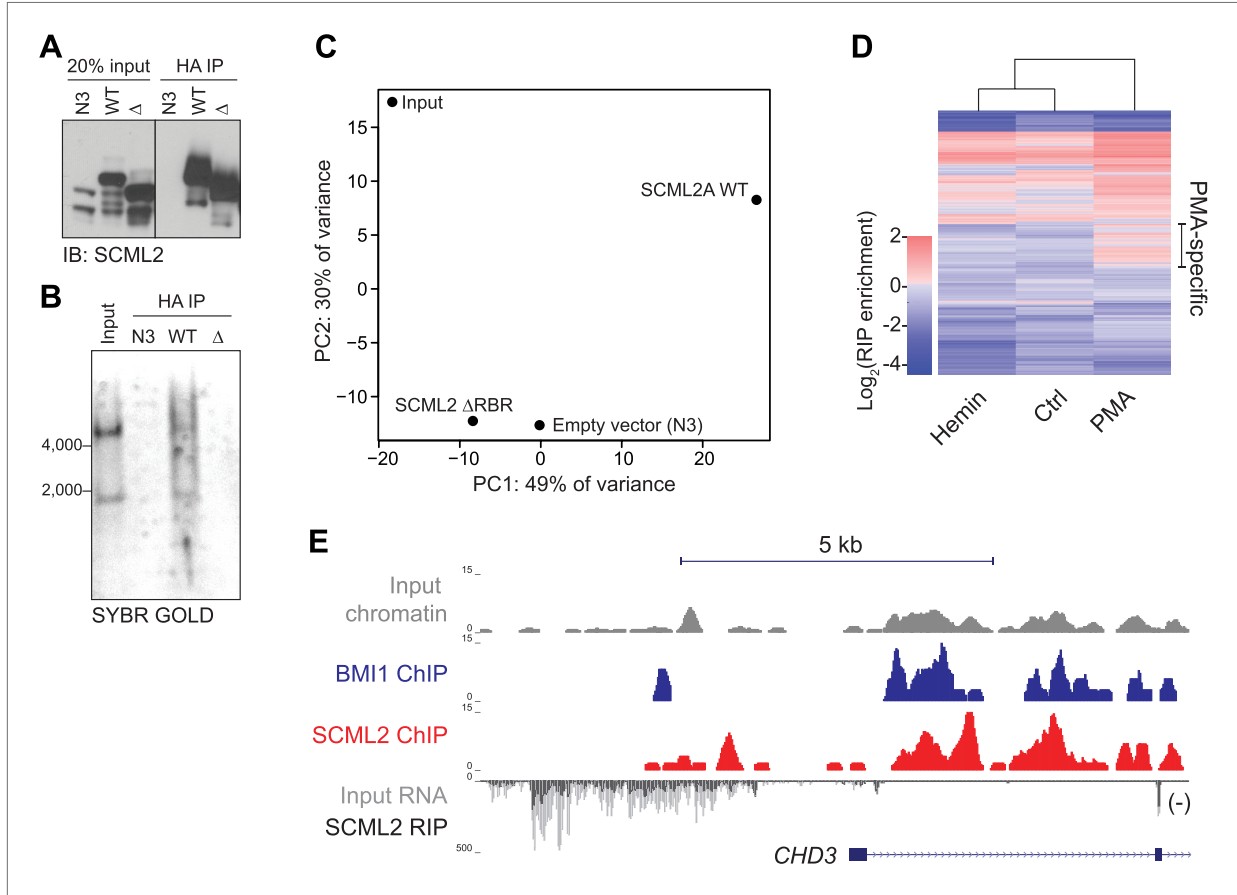

**Figure 6**. RIP-seq for SCML2 in 293T-REx and K562 cells. (**A** and **B**) HA IP from 293T-REx expressing the empty tag alone (N3), N3–SCML2A WT (WT) or N3–SCML2A$_{ΔRBR}$ (Δ). After washing, the bound fraction was divided into two parts; one was analyzed by western blot (**A**) and one by denaturing RNA electrophoresis (**B**). (**C**) Principal component analysis of the lincRNA abundances for the four samples. (**D**) RIP enrichment heatmap for 668 lincRNAs in K562 stimulated to differentiate with hemin or PMA. Colors indicate log$_2$ enrichment in SCML2 RIP-seq compared to the respective inputs. A set of lincRNAs specifically associated with SCML2 in PMA-treated K562 cells is indicated. (**E**) ChIP and RIP profiles at the CHD3 promoter. Normalized read densities are shown for input chromatin (gray), BMI1 ChIP (blue), SCML2 ChIP (red), input RNA (dark gray), and SCML2 RIP (light gray). For simplicity, only the (−) strand of the RNA is shown (antisense to the CHD3 transcript).

The following figure supplements are available for figure 6:

**Figure supplement 1**. Examples of SCML2-associated RNAs.

differentiation caused a measurable shift in the pool of lincRNAs associated with SCML2 (*Figure 6D*), suggesting that in addition to intrinsic binding preferences, cellular context may also determine which RNAs bind to SCML2 and, in turn, how its function and distribution on chromatin is affected by these interactions. This phenomenon was not restricted to lincRNAs. RIP for SCML2 also recovered a variety of mRNAs, some of which were specific for certain states (*Figure 6—figure supplement 1A*).

We also found several examples of unannotated ncRNAs that were enriched in SCML2 RIPs compared to input RNA and were transcribed near SCML2 ERs (*Figure 6E*, *Figure 6—figure supplement 1B–E*), which is consistent with the notion that some ncRNAs interacting with SCML2 might act in *cis*. We found different types of ncRNAs that bound SCML2 and originated near SCML2 target sites on chromatin: (1) divergently transcribed antisense ncRNAs (*Figure 6E*, *Figure 6—figure supplement 1B,D*); (2) ncRNAs that originated downstream of the TSS, were transcribed in the antisense direction, and overlapped with portions of the coding transcript (*Figure 6—figure supplement 1C*); and (3) ncRNAs transcribed upstream of the TSS but from the same strand as the downstream coding gene (*Figure 6—figure supplement 1E*).

Thus, SCML2 associates with annotated and unannotated lncRNAs via the RBR in vivo. These interactions are specific and they are also dynamic, likely reflecting the need to adjust SCML2 localization or regulation as the cells transition from one transcriptional and/or epigenetic state to another.

## SCML2 contributes to gene repression independently of BMI1

Having observed that SCML2 is sufficient to repress a reporter gene after artificial tethering to chromatin (*Figure 4C*), we tested whether its presence is also necessary for the repression of endogenous target genes. To this end, we depleted endogenous SCML2 by siRNA-mediated knockdown for 3 or 5 days with two different siRNAs (*Figure 7A*). We generated genome-wide expression profiles by RNA-seq in two biological replicates and combined all data sets to identify differentially expressed genes (DEGs). Using a loose cutoff of $p < 0.2$, we identified a total of 530 DEGs of which a significant fraction overlapped ($p = 1.4 \times 10^{-4}$, hypergeometric distribution) with the set of genes associated with an SCML2 ER (*Figure 7B*). In keeping with the postulated repressive function of SCML2, a majority (341) of these genes were upregulated (*Supplementary file 1C*) and this subset accounted for ~86% of the DEGs associated with SCML2 occupancy on chromatin (*Figure 7C*). These observations remained valid when we restricted our analysis to the smaller set of genes with an SCML2 ER within 10 kb from the TSS and only considered DEGs with a p-value $<0.05$ (*Figure 7—figure supplement 1*), although the small number of genes obtained with these more stringent filters limited the power of our statistical analysis.

Given that SCML2A can recruit BMI1, and therefore PRC1, to an artificial reporter and that PRC1 is generally thought as the effector of *Polycomb*-mediated repression (*Simon and Kingston, 2009*), we hypothesized that the changes in transcriptional activity observed above might be secondary to a loss of BMI1 from SCML2-bound genes. Unexpectedly, this was not the case. In fact, the levels of BMI1 on chromatin were unaffected even when we analyzed only the 520 sites that lost SCML2 occupancy after *SCML2* knockdown (*Figure 7D,E*). Together with the observation that SCML2B retains the ability to repress a reporter independent of BMI1 recruitment (*Figure 4C*), these findings reinforce the notion that SCML2 represses transcription by itself, and that, in its absence, PRC1 alone is not sufficient to maintain repression at all target genes.

## Discussion

Proteins belonging to the *Polycomb* and *trithorax* group are epigenetic regulators and maintain specialized chromatin structures that are refractory or permissive to transcription, respectively (*Schuettengruber et al., 2007*). Unlike conventional transcription factors, these proteins do not contain a DNA binding domain that directs them to targets on chromatin; instead, there is accumulating evidence that recruitment and regulation of PcG proteins are coordinated by a number of molecular pathways that act in a combinatorial fashion to repress specific genes in different cell types (*Simon and Kingston, 2013*).

*Drosophila* SCM and its human homologs, including SCML2, associate with the PRC1 complex (*Shao et al., 1999*; *Levine et al., 2002*; *Gao et al., 2012*), but their role in PRC1 function has remained elusive. Here, we have unveiled two mechanisms by which SCML2A contributes to PRC1 function: (1) SCML2 and PRC1 cooperate in chromatin binding, as evidenced by loss of BMI1 upon overexpression of SCML2A$_{\Delta RBR}$, recruitment of BMI1 by SCML2A at artificial tethering sites, and the weak association

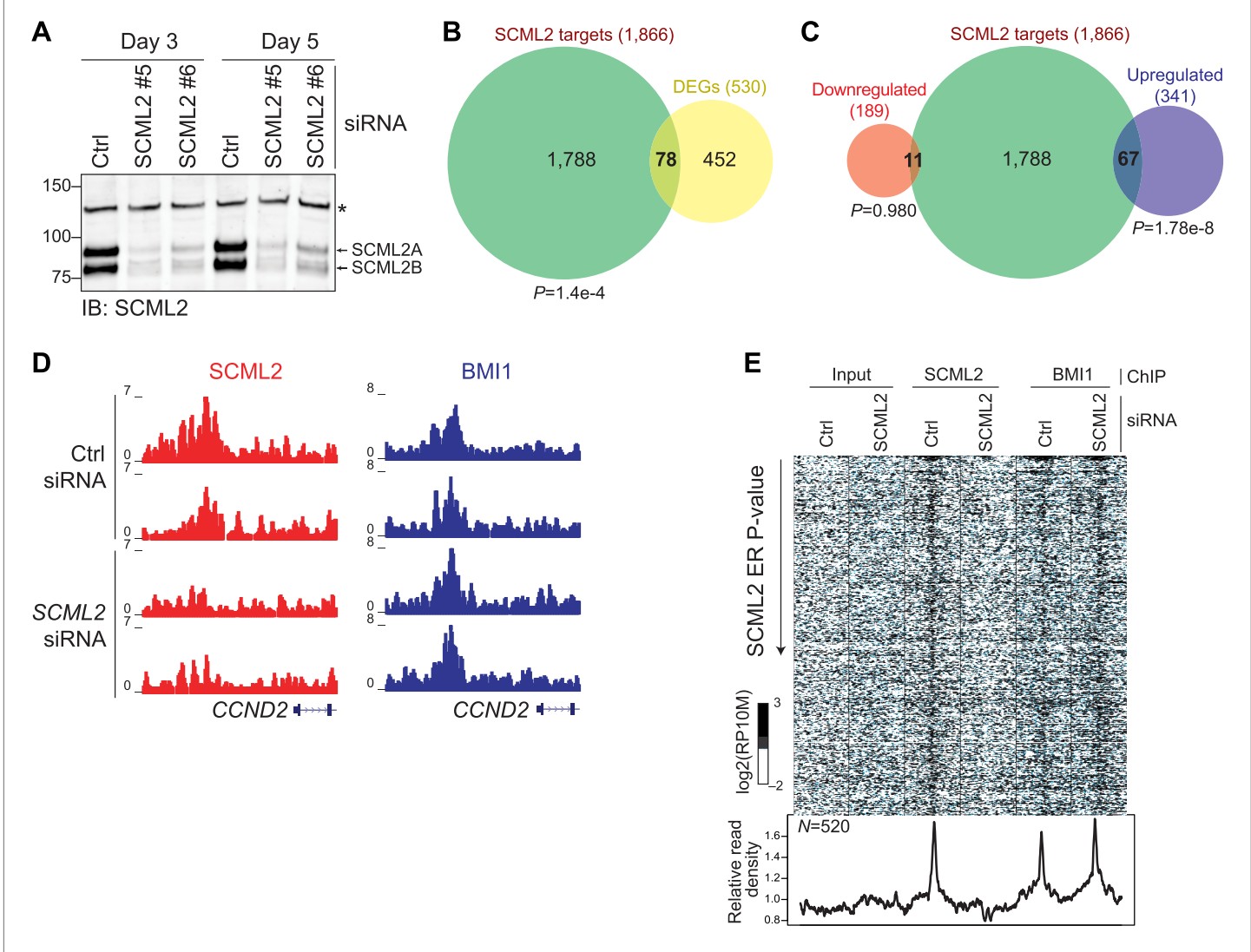

**Figure 7**. Functional consequences of SCML2 depletion. (**A**) Western blot for SCML2 in 293T-REx treated for 3 or 5 days with control siRNAs (Ctrl) or two different siRNAs against *SCML2* (#5 and #6). The band marked with an asterisk corresponds to a cross-reacting protein and serves as loading control. (**B**) Overlap of all DEGs with p<0.2 and genes closest to an SCML2 ER. (**C**) Same as (**B**) but DEGs were divided in downregulated and upregulated. (**D**) Normalized ChIP-seq read densities for SCML2 (red) and BMI1 (blue) in two biological replicates after treating 293T-REx with control (Ctrl) or *SCML2* siRNAs. (**E**) Heatmap for all SCML2 ERs showing decreased occupancy (log$_2$ (SCML2$_{ctrl}$/SCML2$_{KD}$) > 0.5) after *SCML2* KD (n = 520). Profiles for input chromatin, SCML2 and BMI1 ChIP-seq are shown. Read densities were normalized and log-converted. Windows span 2.5 kb on each side of the ER summit and were divided into 25 bp bins.

The following figure supplements are available for figure 7:

**Figure supplement 1**. Additional RNA-seq analyses.

of SCML2B to chromatin; and (2) SCML2 enforces repression at some target sites independent of the presence of PRC1. In addition, our results show that RNA binding also plays a role in SCML2 function, as it is required for its localization to chromatin.

In *Drosophila*, the interaction of SCM and PRC1 relies on the C-terminal SPM domain (**Peterson et al., 2004**) and our results show that a similar mechanism is conserved in humans. Genome-wide analyses revealed an extensive overlap of SCML2A target sites with those of BMI1 (**Figure 4**), suggesting that this interaction is relevant in a chromatin context. Despite the considerable overlap some sites were occupied by SCML2 or BMI1 alone. The former might reflect the complexity of PRC1 subtypes in mammals and the fact that SCML2 is also part of PRC1.2, which does not contain BMI1 (**Gao et al., 2012**); the

latter is likely due to the higher sensitivity of the BMI1 antibody, although we cannot exclude that some BMI1 target sites are devoid of SCML2. Together with genetic evidence in *Drosophila* (*Simon et al., 1992*) and mice (*Takada et al., 2007*), these observations firmly establish SCML2 as an important player in *Polycomb*-mediated epigenetic regulation. Consistent with this, the SCML2B isoform, which lacks the SPM domain, does not bind to BMI1 and is unable to recruit PRC1 to chromatin (*Figure 4—figure supplement 1B*); however, when artificially tethered to a chromatinized reporter, SCML2B is still capable to repress it (*Figure 4C*), suggesting that SCML2 (and possibly other SCM homologs) might directly enforce gene repression. This notion is further supported by the observation that knockdown for *SCML2* causes derepression of target genes without affecting BMI1 levels (*Figure 7*). This PRC1-independent silencing may be mediated by the MBT domain, which binds to methylated histone tails (*Grimm et al., 2007*; *Santiveri et al., 2008*) and, in the case of L3MBTL1, represses transcription via chromatin compaction (*Trojer et al., 2007*). In fact, the MBT domain of SCM in *Drosophila* is important for repression (*Grimm et al., 2007*), and our results show that the MBT domain contributes to the chromatin localization of SCML2 (*Figure 3D*).

In vitro binding assays and domain mapping experiments revealed the existence of a previously unknown RBR within SCML2. Despite the fact that the 74-residue fragment containing the RBR is both necessary and sufficient for RNA binding in vitro (*Figures 1 and 2*), prediction algorithms failed to detect known protein folds or the propensity to form secondary structures in this region, suggesting that, in isolation, the RBR remains unstructured. This is not entirely unexpected, considering that induced fit is a common feature of RNA–protein interactions (*Williamson, 2000*). For example, the fragile X syndrome protein FMR1/FMRP contains an arginine-rich stretch that is disordered in solution but becomes ordered in presence of an RNA ligand (*Phan et al., 2011*). Although the RBR was dispensable for the interaction with PRC1, its disruption resulted not only in a delocalization of SCML2A from chromatin (*Figure 3*), but also in decreased PRC1 occupancy at several target genes (*Figure 5*). This observation is intriguing in view of the fact that short-term *SCML2* knockdown did not affect BMI1 localization, and suggests that nucleoplasmic SCML2$_{\Delta RBR}$, being still capable of interacting with BMI1 through the SPM domain, might function in a dominant negative fashion, titrating PRC1 away from chromatin.

RIP-seq identified several species of RNAs that associate with SCML2, including annotated lincRNAs, unannotated RNAs, and protein-coding mRNAs. Despite promiscuous RNA binding in vitro, SCML2–RNA interactions appeared to be specific in vivo, as demonstrated by the following observations: (1) the profile of lincRNAs enriched in the SCML2A WT RIP-seq was unique and distinct from input and control samples (*Figure 6C*); (2) the lincRNAs associated with SCML2 changed upon cellular differentiation (*Figure 6D*); and (3) several divergently transcribed ncRNAs were enriched in SCML2 RIPs although the coding mRNAs originating from the same locus were not (*Figure 6E*, *Figure 6—figure supplement 1B–E*). The latter also demonstrates that physical proximity on chromatin alone was not the main determinant of SCML2–RNA interactions. Furthermore, the RBR is likely required to mediate the interactions with these RNAs, given that immunoprecipitation of the ΔRBR mutant recovered less RNA than the WT (*Figure 6B*) and that the few detected ΔRBR-associated RNAs were undistinguishable from those obtained in empty tag-expressing cells (*Figure 6C*). We propose that the RBR of SCML2A contributes to its recruitment either by interacting with *trans*-acting RNAs transcribed from distant intergenic loci, or by recognizing *cis*-acting ncRNAs. The latter may include divergent and overlapping antisense ncRNAs that originate downstream of the TSS, some of which have a demonstrated role in gene regulation (*Faghihi and Wahlestedt, 2009*; *Modarresi et al., 2012*), or promoter-associated ncRNAs that originate near the TSS of coding genes and are transcribed from the same strand (*Kapranov et al., 2007*), similar to the ncRNAs reported to bind PRC2 via SUZ12 (*Kanhere et al., 2010*). We note that despite the examples shown (*Figure 6E*, *Figure 6—figure supplement 1B–E*) and several others, we could not detect a significant spatial association between the RIP-seq and ChIP-seq signals at the genome-wide level (data not shown). This might be due to the fact that only a subset of the SCML2-associated ncRNAs act in *cis* or to the intrinsically noisy nature of the RIP-seq signal. Future studies utilizing improved technologies and/or different cellular and genetic tools will be needed to fully characterize the set of SCML2-associated ncRNAs and their mechanism of action.

Using different experimental approaches, we have dissected the molecular interplay of SCML2 and PRC1. We have identified a novel RBR in SCML2 that contributes to its chromatin localization. Along with the recognition of methylated lysines by the MBT domain, the RNA-binding activity of the RBR is

required to recruit or stabilize SCML2 on chromatin. We propose that SCML2A is then retained at target genes through the interaction with PRC1 and that, in turn, SCML2A contributes to stabilizing PRC1. We also report that, at a subset of genes, SCML2 appears to enforce repression directly, independent of PRC1 recruitment, via mechanisms that remain to be elucidated.

The identification of SCML2 as a PcG protein that binds RNA and contributes to repression of *Polycomb* targets will help shed new light on the interplay between ncRNAs and epigenetic regulation and on the combinatorial factors that drive the specificity of the diverse PcG complexes in different cellular contexts.

## Materials and methods

### Cell culture, transfections, and fractionation

HeLa (ATCC), HelaS3 (ATCC), K562 (ATCC), and 293T-REx (Invitrogen, Carlsbad, CA) cells were grown in DMEM with 10% FBS, penicillin (100 IU/ml), streptomycin (100 μg/ml), and L-glutamine (300 μg/ml). Cytosolic and nuclear extracts were prepared as previously described (*Lecona et al., 2008*). The chromatin fraction was either extracted with 8 M urea and 1% Chaps in 50 mM Tris pH 7.9 or solubilized with ammonium sulfate.

Stable transfection of 293T-REx and 293T-REx-luc cells was carried out with polyethilenimine (PEI). The cells were seeded and grown overnight in 10-cm diameter dishes. The next day, 5 μg of plasmid were diluted in 250 μl of 150 mM, and 30 μl of PEI were diluted in 250 μl of 150 mM NaCl. Diluted plasmids and PEI were mixed and incubated for 15 min at room temperature. The mixture was added to the cells and incubated overnight. Next, fresh medium was added to the cells, and they were grown for one additional day. The next day 5 μg/ml blasticidin, 2 μg/ml puromycin (InvivoGen, San Diego, CA), 100 μg Zeocin (Invitrogen), or 1 mg/ml G418 (Sigma-Aldrich, St. Louis, MO) were added individually or in combination, according to the resistance encoded by the plasmid(s) to select resistant clones. Expression of recombinant proteins or shRNAs was induced by adding different amounts of doxycycline for 24 or 48 hr. Lines expressing N3–SCML2A WT, ΔRBR, or empty N3 control were established in two steps. First 293T-REx were transfected with pTRIPZ-V2THS_69827 (Thermo Fisher Scientific, Waltham, MA), and clones were selected with puromycin. The clone displaying the best knockdown efficiency was chosen for further processing and named 'shL2'. The shL2 clone was then transfected with pINTO-N3, pINTO-N3::SCML2A$_{WT}$ or pINTO-N3::SCML2A$_{ΔRBR}$ and sub-clones harboring both the shRNA and the transgene were selected with puromycin and Zeocin and expanded to yield the final lines utilized for the ChIP-seq and RIP-seq experiments.

Transient transfections of HeLa cells with pINTO-GFP plasmids with the different versions of SCML2A and SCML2B were performed with polyfect (Qiagen, Venlo, Netherlands), following manufacturer's instructions. Transfection of the siRNA for human SCML2 (#5 5′-CCAAACGATCTCCTCAGCAAA, #6 5′-CAGTATGTATTGCTACGGTTA) was performed using lipofectamine RNAimax (Invitrogen) according to manufacturer's instructions.

In some experiments K562 were treated for 48 hr with 160 nM PMA in DMSO (Sigma), 100 μM hemin (Sigma-Aldrich), or vehicle (DMSO) alone as control.

### Plasmids and sequences

The construction of the backbones for pcDNA4/TO-GAL4 and pINTO plasmids has been described previously (*Vaquero et al., 2004*; *Gao et al., 2012*). *SCML2A* and *SCML2B* were cloned from human cDNA; truncations and deletions were obtained by PCR.

Details on the oligonucleotides used for this study are reported in *Supplementary file 1D*.

### Antibodies

Rabbit antibody against SCML2 was generated using an antigen consisting of a central region of SCML2 fused to GST and affinity-purified using the same antigen fused to a 6xHis tag. The following antibodies were used for IP and ChIP experiments: HA (ab9110; Abcam, Cambridge, England), BMI1 (A301-694A; Bethyl, Montgomery, TX), RING1B (A302-869A; Bethyl), GAL4 (06-262; Millipore). Antibodies against GST (sc-33613; Santa Cruz Biotechnology, Santa Cruz, CA) or RBP1 (Reinberg lab) were used for Western Blots.

### RNA immunoprecipitation

Nuclear extracts were obtained using an established protocol (*Dignam et al., 1983*) with minor modifications to minimize RNAse activity. Briefly, cells were washed with PBS and with Buffer A (10 mM Tris

pH 7.9$_{4°C}$, 1.5 mM MgCl$_2$, 10 mM KCl, protease inhibitors, phosphatase inhibitors) and lysed in Buffer A with 0.2% IGEPAL CA-630 for 5 min on ice. Nuclei were isolated by centrifugation at 2,500×$g$ for 5 min and lysed in Buffer C (20 mM Tris pH 7.9$_{4°C}$, 25% glycerol, 400 mM NaCl, 1.5 mM MgCl$_2$, 10 mM EDTA, 0.4 µ/µl murine RNAse inhibitor, protease inhibitors, phosphatase inhibitors) for 30 min at 4°C. Lysates were cleared at 20,000×$g$ for 30 min.

For IP, lysates were diluted to 1 mg/ml in RIP buffer (20 mM Tris pH 7.9$_{4°C}$, 200 mM KCl, 0.05% IGEPAL CA-630, 10 mM EDTA), cleared by centrifugation at 20,000×$g$ for 10 min, and incubated with a depleting amount of antibody (predetermined with titration experiments) for 3 hr at 4°C. Immunocomplexes were recovered by incubating for 1 hr at 4°C with 7 µl of protein G-coupled Dynabeads (Invitrogen) per µg of antibody used. Beads were washed in RIP-W buffer (RIP buffer without EDTA and with 1 mM MgCl$_2$) twice and incubated with 2 U of TURBO DNase (Life Technologies, Carlsbad, CA) in 20 µl RIP-W buffer for 10 min at room temperature, to eliminate potential bridging effects of protein–DNA and RNA–DNA interactions. After two additional washes in RIP-W buffer RNA was eluted from the beads with TRIzol and collected by precipitation with isopropanol. Residual DNA was removed with TURBO DNAse for 20 min at 37°C.

## Recombinant protein expression and purification

GST fusion proteins were expressed in BL21(DE3) cells for 16 hr at 16°C and purified using a glutathione–sepharose column. The column was washed extensively with 50 mM Tris pH 7.5, 50 mM NaCl, and the proteins were eluted in the presence of 10 mM glutathione. The purified proteins were dialyzed against 50 mM tris pH 7.5, 50 mM NaCl and 10% glycerol.

## In vitro RNA binding assays

Single-stranded DNA fragments were commercially synthesized (IDT) and annealed to generate double-stranded species. Single-stranded RNA were in vitro transcribed with the HiScribe kit (New England Biolabs, Ipswich, MA) and purified by size exclusion on mini Quick Spin columns (Roche, Penzberg, Germany) followed by TRIzol (Invitrogen) extraction and isopropanol precipitation. *HOTAIR* RNA fragments were obtained from a construct kindly gifted by H Chang (Stanford).

For the binding assays, GST-tagged proteins and protein fragments were incubated with RNA or DNA in 100 µl RIP-W buffer (20 mM Tris pH 7.9$_{4°C}$, 1 mM MgCl$_2$, 0.05% IGEPAL CA-630) with the addition of 2 U/µl murine RNAse inhibitor (New England Biolabs) for 1 hr at 4°C. Protein–RNA complexes were precipitated by adding 10–20 µl of glutathione-coupled beads (Qiagen) for 30 min at 4°C. After three washes with RIP-W buffer, proteins were eluted from the beads with Laemmli sample buffer and nucleic acid with TRIzol or phenol/chloroform/isoamyl alcohol. RNA and DNA were resolved on polyacrylamide/urea gels and visualized with SYBR gold (Invitrogen).

## Electrophoretic mobility shift assays (EMSAs)

The GST-tagged RBR corresponding to residues 256–330 of human SCML2 was purified as described above. In some cases, the GST tag was cleaved off with prescission protease (GE Healthcare, Little Chalfont, UK) and residual uncleaved protein removed with a GST column. The 1–300 fragment of HOTAIR and a 200-nts construct spanning the 601 nucleosome positioning sequence were obtained by in vitro transcription. In vitro-transcribed HOTAIR was end-labeled with [$^{32}$P]-γ-ATP and purified by phenol–chloroform extraction. RNA was refolded prior to EMSA assays by heating at 95°C for 5 min in BTE buffer (10 mM bis-tris pH 6.7, 1 mM EDTA) then incubating on ice and room temperature for 5 min each. Double-stranded DNA encoding *HOTAIR* 1–300 was generated by PCR. Nucleosome core particles were generated by salt dialysis of recombinant histone octamers and a 147-bp fragment corresponding to the 601 nucleosome positioning sequence, as described (*Voigt et al., 2012*). Dialysis was performed as a gradient of 2–0.4 M NaCl overnight at 4°C, followed by dialysis into TE buffer. Recombinant wild-type *Xenopus laevis* core histones were purified from *E. coli*.

Protein and nucleic acid were co-incubated for 30 min on ice in 10 mM HEPES pH 7.9, 100 mM KCl, 0.5 mM EDTA, 5% glycerol, 0.25 mg/ml BSA, and 400 U/ml murine RNAse inhibitor (New England Biolabs). 50 ng of RNA, double-stranded DNA, or nucleosomes were incubated with 70–140 ng of GST or GST-fused RBR, or 17–35 ng of RBR without GST in 15-µl binding reactions. Samples were loaded onto 4% acrylamide mini gels pre-run for 1 hr at 100 V in 0.5x TGE (12.5 mM Tris pH 8.8, 95 mM glycine, 0.5 mM EDTA). Electrophoresis was performed for 2 hr at 100 V at 4°C. Gels were stained with SYBR gold (Invitrogen) and documented with a LAS-4000 imaging system (GE Healthcare). Radiolabeled RNA was detected using a phosphoimaging screen (GE Healthcare).

## Nucleosome pull-downs

Nucleosome core particles were reconstituted on biotinylated 147-bp 601 DNA. Streptavidin agarose beads (Millipore, Billerica, MA) were blocked with 1% BSA in pull-down buffer (10 mM HEPES pH 7.9, 100 mM KCl, 0.5 mM EDTA, 5% glycerol, 0.01% NP-40, 400 U/ml murine RNAse inhibitor) for 4 hr at 4°C. Binding reactions containing 1 µg of nucleosomes, 1 µg of RBR, and varying amounts of *HOTAIR* RNA (100–400 ng) were incubated on ice for 30 min and then with the blocked beads for 3 hr at 4°C. After two washes, beads from each reaction were divided for western blot and RNA analysis. RNA was purified with TRIzol (Invitrogen) and analyzed by non-denaturing acrylamide gel electrophoresis. Proteins were eluted by boiling in 1x SDS loading buffer and analyzed by Western blot with histone H3 and GST antibodies.

## Luciferase assay

$5 \times 10^4$ 293T-REx-luc cells stably transfected with pcDNA4/TO::GAL4-SCML2A or SCML2B were seeded in 24-well plates. The cells were incubated for 24 hr in the presence of different doxycycline concentrations or the equivalent amount of ethanol as a control. After washing twice with PBS, the cells were resuspended in 100 mM potassium phosphate pH 7.8 containing 0.2% triton X-100, and lysed for 15 min at 4°C with agitation. Cell debris was removed by centrifugation at 20,000×*g* for 10 min, and the supernatant was assayed for luciferase activity using a luciferase assay system according to manufacturer's instructions (Promega, Madison, WI).

## Fluorescence microscopy

48 hr after transfection, cells were washed three times and fixed with 4% paraformaldehyde in PBS for 10 min at room temperature. After washing twice with PBS and twice with water, the cells were mounted using SlowFade with DAPI (Invitrogen) and imaged using an Axiovert 200 M inverted microscope (Zeiss, Oberkochen, Germany).

## Purification of TwinStrepTag-tagged proteins

30 mg of solubilized nuclear pellet from 293T-REx expressing FS-SCML2A or SCML2B, wild-type or ΔRBR were diluted in 10 ml of 50 mM Tris pH 7.5 with 200 mM NaCl and incubated with 600 µl of Strep-Tactin resin (IBA, Göttingen, Germany) for 1 hr at 4°C with constant agitation. The resin was allowed to settle and the flow-through was collected. The columns were washed with 10 ml of 50 mM Tris pH 7.5, 300 mM NaCl, and 0.1% Igepal CA630 (Sigma), and then with 4 ml of 50 mM Tris pH 7.5, 200 mM NaCl. The proteins bound to the resin were eluted in denaturing loading buffer and analyzed by mass spectrometry.

## Chromatin immunoprecipitation

Cells were fixed in DMEM containing 10 mM Hepes, pH 7.6, 1% formaldehyde, 15 mM NaCl, 0.15 mM EDTA and 0.075 mM EGTA, for 10 min at room temperature. The reaction was quenched with 0.125 M glycine for 5 min at room temperature. After washing with PBS, cells were collected, and the pellet was lysed in 50 mM Hepes, 140 mM NaCl, 1 mM EDTA, 10% glycerol, 0.5% Igepal CA630, and 0.25% Triton X-100. After centrifugation, isolated nuclei were washed once in 10 mM Tris pH 8.0, 200 mM NaCl, 1 mM EDTA and 0.5 mM EGTA, then resuspended in 10 mM Tris pH 8.0, 1 mM EDTA, 0.5 mM EGTA and 0.5% N-Lauryl Sarcosine, and sonicated using a Diagenode Bioruptor to an average chromatin size of 200 bp. Chromatin was diluted to 150–200 µg/300 µl in 10 mM Tris pH 8.0, 1 mM EDTA, 600 mM NaCl and 3% Triton X-100, the antibodies were added (4–8 µg) and incubated at 4°C overnight. Protein G Dynabeads (Invitrogen) were blocked with BSA, and 20–40 µl of beads were added to each IP. After 1 hr of incubation at 4°C, the beads were washed five times with 750 µl of 25 mM Hepes pH 7.6, 1 mM EDTA, 0.1% N-Lauryl Sarcosine, 1% NP-40, and 0.5 M LiCl. After a final wash with 750 µl of 10 mM Tris pH 8.0, 1 mM EDTA and 50 mM NaCl, the beads were resuspended in 100 mM sodium bicarbonate, 200 mM NaCl, and 1% SDS. 1 µg of proteinase K was added to each sample, the chromatin was incubated 15 min at room temperature and the cross-linking was reversed at 65°C for 16 hr. 10% of the input was treated in parallel. The DNA was extracted using a PCR purification kit (Qiagen).

## Library construction

Libraries for ChIP-seq were prepared according to manufacturer's instructions (Illumina, San Diego, CA) and as described (*Asp et al., 2011*). Briefly, IP'ed DNA (~5 ng) was end-repaired using End-It Repair Kit (Epicenter, Madison, WI), tailed with deoxyadenine using Klenow exo⁻ (New England Biolabs),

and ligated to custom adapters with LigaFast (Promega). Fragments of 300 ± 50 bp were size-selected and subjected to ligation-mediated PCR amplification using Phusion DNA polymerase (New England Biolabs). Libraries were quantified by qPCR using primers annealing to the adapter sequence and sequenced at a concentration of 7 pM on an Illumina Genome Analyzer IIx or 10 pM on an Illumina HiSeq. In some cases barcoding was utilized for multiplexing.

For RIP-seq libraries, polyA+ RNA was isolated using Dynabeads Oligo(dT)$_{25}$ (Invitrogen) beads and constructed into strand-specific libraries using the dUTP method (*Parkhomchuk et al., 2009*). Once dUTP-marked double-stranded cDNA was obtained, the remaining library construction steps followed the same protocol as described above for ChIP-seq libraries.

### ChIP-seq, RNA-seq, and RIP-seq analysis

ChIP-seq analysis was performed as described before with modifications (*Gao et al., 2012*). Sequenced reads from ChIP-seq experiments were mapped with Bowtie using parameters –v2 –m4 (*Langmead et al., 2009*). Normalized genome-wide read densities were visualized on the UCSC genome browser (http://genome.ucsc.edu). Enriched regions (ERs) were identified using MACS 1.40rc2 (*Zhang et al., 2008*) using default parameters requiring at least 10 reads per ER and an unadjusted p-value <10$^{-5}$. GO analyses were performed with DAVID (*Huang da et al., 2009*) and GREAT (*McLean et al., 2010*). Heatmaps were generated in R.

RIP-seq and RNA-seq reads were mapped to the reference genome with Bowtie using parameters –v2 –m40. Normalized genome-wide read densities for each strand separately were visualized on the UCSC genome browser. Reads were assigned to genes using DEGseq (*Wang et al., 2010a*) and either the ENSEMBL annotation or the lincRNA catalog (*Cabili et al., 2011*) made available by the Broad Institute. DEG identification was performed with the R package DEseq (*Anders and Huber, 2010*).

### Sequencing data

All sequencing data is available in the NCBI GEO as SuperSeries GSE38275.

## Acknowledgements

We wish to thank Moran Cabili and John Rinn for their help with lincRNA analyses. This work was supported by grants from the National Institute of Health (GM-64844 and R37-37120) and the Howard Hughes Medical Institute (to DR). RB was supported by a Helen Hay Whitney Foundation postdoctoral fellowship and by the Helen L and Martin S Kimmel Center for Stem Cell Biology postdoctoral fellow award. EL was supported by an International Outgoing Fellowship from the European Commission (Marie-Curie Actions, FP7). PV was supported by a fellowship from the Deutsche Akademie der Naturforscher Leopoldina (LPDS 2009-5) and subsequently by a fellowship from the Empire State Training Program in Stem Cell Research (NYSTEM, contract no. C026880). FP and YK were supported by a grant from the National Cancer Institute and National Institute of Health (CA-16359 to YK).

## Additional information

### Competing interests

DR: Reviewing editor, *eLife*. The authors declare that no competing interests exist.

### Funding

| Funder | Grant reference number | Author |
| --- | --- | --- |
| National Institutes of Health (NIH) | GM-64844, R37-37120 | Danny Reinberg |
| Howard Hughes Medical Institute (HHMI) | | Danny Reinberg |
| National Institutes of Health (NIH) | CA-16359 | Fabio Parisi, Yuval Kluger |

The funders had no role in study design, data collection and interpretation, or the decision to submit the work for publication.

### Author contributions

RB, EL, Conception and design, Acquisition of data, Analysis and interpretation of data, Drafting or revising the article; VN, PV, Acquisition of data, Analysis and interpretation of data; FP, YK, Analysis

and interpretation of data, Drafting or revising the article; DR, Conception and design, Drafting or revising the article

## Additional files

### Supplementary file

• Supplementary file 1. (A) GO enrichment analysis for SCML2 targets in 293T-Rex. (B) SCML2 target genes in 293T-REx. (C) Differentially expressed genes after SCML2 KD. (D) Oligonucleotide sequences.

### Major dataset

The following dataset was generated:

| Author(s) | Year | Dataset title | Dataset ID and/or URL | Database, license, and accessibility information |
|---|---|---|---|---|
| Bonasio R, Lecona E, Reinberg D | 2014 | ChIP-seq and RIP-seq for SCML2 and SCML2 mutants in 293T-REx and K562 cells | http://www.ncbi.nlm.nih.gov/geo/query/acc.cgi?acc=GSE38275 | Publicly available at NCBI Gene Expression Omnibus. |

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
