## [Decision Letter]

Your manuscript titled "Interactions with RNA direct SCML2 to chromatin where it represses PRC1 target genes" was reviewed by a Senior Editor, three experts in the field, and by a member of the Board of Reviewing Editors. After full discussion of the study and the reviews, we are happy to report that the reviewers and the BRE member are enthusiastic about considering a revised version of the paper following the implementation of their suggestions.

Please address the following points in your revised manuscript:

1) Do the loci that produce RNA that associates with SCML2A correspond to those sites of PRC1/SCML2A co-occupancy? The reviewers have suggested that you should address this point with mRNAs and lncRNAs in the same cell type where RIPs were performed.

2) The reviewers were concerned that in Figure 1, RBR mainly co-precipitates with ribosomal RNAs. This point needs further clarification in the text of the manuscript for our readerships. Could you also please clarify if there is a co-IP of PRC2 and SCML2A?

3) Since SCML2 associates with the PRC1 complex, then the genome-wide distribution of the peaks of both proteins should largely overlap. All the analyses presented here refer to common targets, rather than peak distributions. This observation needs to be further clarified in the revised manuscript.

4) In the Results section you have stated: "...suggesting that RNA-protein interactions participate in the recruitment or stabilization of SCML2 on chromatin..." should be toned down as 'might participate in'.

5) In the ChIP-seq experiments with the WT HA-tagged SCML2 and the "delta RBR" mutant in cells with endogenous SCML2 kd - it is unclear from the Western on Figure 4, how much of a knockdown effect we are observing since the levels of SCML2A and SCML2B in a control shRNA or in the parental cell line is not provided; for the same reason, we don't know how much of the HA-tagged proteins we are actually expressing compared to normal endogenous levels; this information is important to interpret the results, therefore, the normal levels of SCML2 in a shRNA control or parental cell line should be provided to be able to compare. Similarly, in the ChIP-seq experiments (Figures 4 and 5), the control with no SCML2-HA-tagged expressed in the SCML2 shRNA conditions is missing.

6) The reviewers suggested that the alignments to whole transcriptome and not just lncRNAs should be performed in the study.

7) Although the reviewers requested demonstration of physiological significance of the findings, upon discussion it was concluded that such information could be the subject of the next study. However, if you have such data already available, we feel that its inclusion would significantly strengthen the manuscript.

---

## [Author Response]

1) Do the loci that produce RNA that associates with SCML2A correspond to those sites of PRC1/SCML2A co-occupancy?

We performed the requested analyses by calculating whether the SCML2 RIP signal near SCML2 ERs was enriched compared to input, empty tag, and ΔRBR RIP. Using all 2,479 SCML2 ERs in 293T-REx, no specific enrichment can be observed (Figure 8). However, the 2,479 sites are likely to include false positive and/or RNA-independent SCML2 targets. We reasoned that the noise in the RIP data could be filtered by focusing on those regions where the WT RIP showed more signal than the ΔRBR RIP. These ERs showed an accumulation of signal in the SCML2 WT RIP but not in the other 2 independent controls (Figure 8). As an additional control, we analyzed the ERs that showed stronger RIP signal in the ΔRBR sample. In these regions the ΔRBR signal was as high as the input and empty vector controls and can be attributed to non-specific interactions (Figure 8).Author response image 1.Enrichment of SCML2-bound RNAs at SCML2 ERs on chromatin.(A) The *cis* enrichment score was calculated by first counting the reads per million sequenced (RP10M) for each RIP sample in 5kb windows spanning the SCML2 ERs, dividing by the RP10M for the same sample in 1,000 simulations of an equal number of randomly selected regions in the genome, and then normalized to the input signal. Bars represent the mean score over all ERs + s.e.m. (B) Same as (A) but restricting the analysis to those SCML2 ERs where the RP10M for the WT RIP was higher than the RP10M for the ΔRBR RIP. (C) Same as (A) but restricting the analysis to the SCML2 ERs filtered out in (B).

In summary, we believe that the examples of adjacent RIP/ChIP enrichment shown (Figure 6—figure supplement 1) might be representative of a small subset of geneswhere the ncRNA bound to SCML2 act in *cis*, or that a more general phenomenon might be masked by the poor signal-to-noise ratio of RIP-seq, a known drawback of this technique. We have added text in the Discussion reporting some of these considerations.

*The reviewers have suggested that you should address this point with mRNAs and lncRNAs in the same cell type where RIPs were performed*.

Most genomic regions with RIP enrichment do not contain gene annotations. For this reason, the analyses mentioned above were performed on raw reads. Annotations for lincRNAs and mRNAs were used for Figure 6 and new Figure 6—figure supplement 1.

*2) The reviewers were concerned that in*
Figure 1*, RBR mainly co-precipitates with ribosomal RNAs. This point needs further clarification in the text of the manuscript for our readerships*.

The experiment in Figure 1 was only meant to identify the region within SCML2 that is responsible for binding to RNA in general and not the relative affinity of different RNAs for SCML2. Because total RNA was used and the most abundant RNA species are rRNAs, they are also the most prominent bands in this gel. It is not unusual for chromatin-associated proteins to have broad binding specificities, at least *in vitro* (11; 19).

We added a comment about this in the revised manuscript.

Could you also please clarify if there is a co-IP of PRC2 and SCML2A?

We do not observe co-IP of PRC2 and SCML2A in native conditions (Figure 9) or after crosslinking with formaldehyde (Figure 9). Kuroda and colleagues recovered SCML2 in their EZH2 crosslinking and affinity purification (Alekseyenko et al., 2014); however, they only detected 2 peptides (compared to 124 for SUZ12). This reflects either an extremely weak interaction, or indirect crosslinking due to co-localization or PRC1 and PRC2 components on chromatin targets. We have included a brief mention of these considerations in the revised manuscript. We can add Figures R2A and R2B to the supplement upon request, but we feel that they would add little to the paper.Author response image 2.Immunoprecipitations for SCML2 and EZH2.(A) IPs were performed in native conditions (200 mM KCl, 0.05% NP-40) with the indicated antibodies (C-20 and 86A are two different commercial antibodies against MLL). Flow-through (FT) and IP fractions were resolved on SDS-PAGE and immunoblotted (IB) for MLL (top), EZH2 (middle), and SCML2 (bottom). goIG, goat IgG control; rbIg, rabbit IgG control. (B) As in (A) but IPs were performed after crosslinking with formaldehyde as for the ChIP protocol. Crosslinks were reversed before SDS-PAGE by boiling for 20’. Note that SCML2A but not SCML2B co-immunoprecipitates with BMI1, as expected.

*3) Since SCML2 associates with the PRC1 complex, then the genome-wide distribution of the peaks of both proteins should largely overlap*.

SCML2 and BMI1 targets overlap extensively in Figure 4. It is rare to observe a full overlap in genome-wide datasets, due to different antibody sensitivity and protein abundance. By increasing the stringency of our peak calling, an even larger fraction of SCML2 ERs overlap with BMI1 ERs (new Figure 4—figure supplement 1). Relevant text has been added (Pg. 9). In addition, BMI1 is part of only one (PRC1.4) of the several types of PRC1 complexes present in mammalian cells (15). Although PRC1.1/3/5/6 do not appear to interact with SCML2, PRC1.2 does (15), and therefore it is possible that some of the SCML2+/BMI1- ERs are bound by PRC1.2. We added these considerations to the revised Discussion.

*All the analyses presented here refer to common targets, rather than peak distributions. This observation needs to be further clarified in the revised manuscript*.

Because the overlap between SCML2 and BMI1 decreases at higher *P*-values (see new Figure 4—figure supplement 1) and to minimize the inclusion of ERs that might be the result of cross-reactivities of the anti-SCML2 antibody, we restricted some of the analyses to a “high-quality” set of ERs comprising the intersection of SCML2, BMI1, and HA ERs (Figure 5, Figure 5—figure supplement 1). However, the meta-ER analyses shown in Figure 5 were performed on the inclusive set of all 2,479 SCML2 ERs with *P*-value < 10-5, and the gene set overlap shown in Figure 7 included all unique target genes associated with these 2,479 ERs (1,866 total; some genes are associated with more than one ER).

The average enrichment in WT vs. ΔRBR decreases as ERs with higher *P*-values are added to the set (Figure 10) suggesting a corresponding increase in the fraction of false positive ERs. Nonetheless the heatmap including all 2,479 ERs (Figure 10) is similar to the one restricted to the high-confidence set (Figure 5). However, because Figure 5 already show the appropriate comparisons using the full set, we think that Figure 10 would not add to the manuscript. We have also added explanatory text to the Results section to clarify which set of ERs was used for which analysis.Author response image 3.Comparison of WT and ΔRBR ChIP-seq on unfiltered SCML2 ERs.(A) Enrichment is defined as the normalized number of reads for the indicated sample in 1 kb regions centered on the ER summit in 293T-REx expressing N3–SCML2 WT minus the reads in the same regions in cells expressing N3–SCML2 ΔRBR. The average enrichment for each set of ERs, as defined by their *P*-value is shown on the *y* axis, starting from the full SCML2 ER set on the right (*N* = 2,479, *P* < 10^-5^), to only ERs with *P* < 10^-26^ on the left of the graph (*N* = 24). (B) Same heatmap as shown in Figure 5, but this time including all 2,479 SCML2 ERs.

*4) In the Results section you have stated: "…suggesting that RNA-protein interactions participate in the recruitment or stabilization of SCML2 on chromatin..." should be toned down as 'might participate in'*.

We edited as requested. We also toned down some of the language in other parts of the manuscript, including the title of this section and the title of Figure 5.

*5) In the ChIP-seq experiments with the WT HA-tagged SCML2 and the "delta RBR" mutant in cells with endogenous SCML2 kd – it is unclear from the Western on*
Figure 4*, how much of a knockdown effect we are observing since the levels of SCML2A and SCML2B in a control shRNA or in the parental cell line is not provided; for the same reason, we don't know how much of the HA-tagged proteins we are actually expressing compared to normal endogenous levels; this information is important to interpret the results, therefore, the normal levels of SCML2 in a shRNA control or parental cell line should be provided to be able to compare*.

To establish the different 293T-REx lines, we first generated a clone harboring an inducible shRNA transgene against *SCML2* (“*shL2*” clone); then, we transfected the different transgenic constructs (N3, N3–SCML2AWT and N3–SCML2AΔRBR) into the *shL2* clone and new clones harboring both the inducible shRNA and the inducible SCML2 constructs were selected. As they all derived from the same *shL2* clone, the extent of knockdown is equivalent in all cell lines and can be gauged by comparing the levels of endogenous (untagged) SCML2 before and after dox in Figure 5—figure supplement 1. This figure also allows the direct comparison between the levels of exogenous and endogenous SCML2, as both types of protein are detected by the SCML2 antibody. Because the shRNA is not induced in absence of dox, we deemed the –dox condition a better control than the parental cell line. Indeed, the levels of SCML2 in the empty vector uninduced line (N3 –dox) line are equivalent to those in the parental 293T-Rex line (Figure 11). We have added this comment in the Results section. We also clarified how the cell lines were derived in the Materials and methods section.Author response image 4.SCML2 expression in parental 293T-REx and N3 cell lines.293T-REx either untreated (-) or treated with 1 µg/ml doxycycline for 48 h were lysed and proteins resolved on SDS-PAGE. Immunoblots for SCML2 (top) and CDK2 (bottom) as a loading control are shown. Note the knockdown of SCML2 in the N3 line +dox and that the levels of SCML2 in N3 cells -dox are equivalent to those in the parental 293T-REx line.

*Similarly, in the ChIP-seq experiments (*Figures 4 and 5*), the control with no SCML2-HA-tagged expressed in the SCML2 shRNA conditions is missing*.

Based on the observation that the inducible shRNA system only resulted in ∼50% knockdown of endogenous SCML2 (Figure 5—figure supplement 1, Figure 11) we did not perform ChIPseq in induced empty vector lines (“no SCML2-HA-tagged in SCML2 shRNA conditions”). The knockdown obtained with siRNAs is more efficient (Figure 7), which makes the siRNA experiment more meaningful. Note that the time during which the cells were exposed to knockdown conditions is comparable. We waited three days after transfection of the siRNAs, while in the shRNA experiments we harvested two days after induction.

Adding the requested ChIP-seq data now would require performing three replicates of ChIP-seq for all factors not only in the empty vector and shRNA-induced conditions, but also in the lines expressing SCML2 WT and ΔRBR, because otherwise quantitative comparisons between different experiments would be difficult to interpret. This would be a very expensive set of experiments and would require more than the 2 months suggested for this revision.

We did perform ChIP-seq for the empty vector line in the absence of dox in replicate #2 and #3, mostly for the purpose of validating the HA antibody, which, as expected, shows no signal these cells. We have included these data in revised Figure 5—figure supplement 1.

*6) The reviewers suggested that the alignments to whole transcriptome and not just lncRNAs should be performed in the study*.

The lincRNA DB (8) was only used for Figure 6. The purpose of those figures was to show that the set of lincRNAs associated with SCML2 changes with cellular state. Indeed SCML2 binds as efficiently to mRNA as it does to lincRNAs by RIP-seq. This is expected based on its behavior *in vitro*. We have added the same analysis for protein-coding mRNAs (Figure 6—figure supplement 1) and some explanatory text.

*7) Although the reviewers requested demonstration of physiological significance of the findings, upon discussion it was concluded that such information could be the subject of the next study. However, if you have such data already available, we feel that its inclusion would significantly strengthen the manuscript*.

This is the subject of an ongoing study still too preliminary to be included in this manuscript.